# Improved CRISPR/Cas9 off-target prediction with DNABERT and epigenetic features

**Kai Kimata[1], Kenji Satou[2]\***

1 Graduate School of Natural Science and Technology, Kanazawa University, Kanazawa, Japan,
2 Faculty of Transdisciplinary Sciences for Innovation, Institute of Transdisciplinary Sciences for Innovation, Kanazawa University, Kanazawa, Japan

\* ken@t.kanazawa-u.ac.jp

## Abstract

CRISPR/Cas9 is a powerful genome editing tool, but its clinical application is hindered by off-target effects. Accurate computational prediction of these unintended edits is crucial for ensuring the safety and efficacy of therapeutic applications. While various deep learning models have been developed, most are trained only on task-specific data, failing to leverage the vast knowledge embedded in entire genomes. To address this limitation, we introduce a novel approach that integrates DNABERT, a deep learning model pre-trained on the human genome, with epigenetic features (H3K4me3, H3K27ac, and ATAC-seq). We conducted a comprehensive benchmark of our model, DNABERT-Epi, against five state-of-the-art methods across seven distinct off-target datasets. Our results demonstrate that the pre-trained DNABERT-based models achieve competitive or even superior performance. Rigorous ablation studies quantitatively confirmed that both genomic pre-training and the integration of epigenetic features are critical factors that significantly enhance predictive accuracy. Furthermore, by applying advanced interpretability techniques (SHAP and Integrated Gradients), we identified the specific epigenetic marks and sequence-level patterns that influence the model's predictions, offering insights into its decision-making process. This study is the first to establish the significant potential of a pre-trained DNA foundation model for CRISPR/Cas9 off-target prediction. Our findings underscore that leveraging both large-scale genomic knowledge and multi-modal data is a key strategy for advancing the development of safer genome editing tools.

## 1. Introduction

The CRISPR (Clustered Regularly Interspaced Short Palindromic Repeats)/Cas9 system has revolutionized the field of biology, providing an unprecedentedly simple and efficient tool for genome editing [1]. Originally discovered as an adaptive immune system in bacteria, its adaptation for targeted DNA cleavage has opened up vast possibilities in genetic engineering, functional genomics, and the development of novel

**Data availability statement:** All relevant data are within the manuscript and its Supporting information files. All source code used in this study is available from GitHub: https://github.com/kimatakai/CRISPR_DNABERT.

**Funding:** The author(s) received no specific funding for this work.

**Competing interests:** The authors have declared that no competing interests exist.

gene therapies for a wide range of human diseases [2–6]. The system's specificity is primarily guided by a 20-nucleotide sequence in the single-guide RNA (sgRNA), which directs the Cas9 nuclease to a complementary target DNA sequence adjacent to a protospacer adjacent motif (PAM) [7–9].

Despite its power and precision, the therapeutic application of CRISPR/Cas9 is hampered by the risk of off-target effects, where the Cas9 nuclease cleaves unintended genomic sites that are similar to the intended target sequence [7–9]. Such unintended edits can lead to deleterious consequences, including the disruption of essential genes or the activation of oncogenes, posing a significant safety concern for clinical applications [10–12]. Therefore, the ability to accurately predict potential off-target sites *in silico* is of paramount importance for designing safe and effective sgRNAs.

In response to this challenge, a multitude of computational methods have been developed, evolving from early scoring algorithms to more sophisticated deep learning models that have demonstrated superior predictive performance [13–15]. In recent years, models based on the Transformer architecture, a cornerstone of natural language processing, have been successfully applied to off-target prediction, with models such as CRISPR-BERT and CrisprBERT showing promising results [16,17]. Concurrently, similar deep learning approaches have been effectively utilized in related bioinformatics classification tasks, such as predicting protein modification sites and identifying regulatory elements [18–22]. However, many existing deep learning models for off-target prediction are trained exclusively on task-specific datasets. This approach overlooks the rich, contextual information embedded within the entire genome. Moreover, while accumulating evidence suggests that epigenetic factors, such as chromatin accessibility, influence Cas9 activity [23,24], the integration of these features into predictive models remains an area of active development.

This study introduces a novel approach to address these limitations, centered on three key contributions. First, we present the first application of a pre-trained DNA foundation model, DNABERT, to the CRISPR/Cas9 off-target prediction task [25]. Unlike models trained from scratch on limited data, DNABERT has been pre-trained on the entire human genome, allowing it to learn the fundamental "language" of DNA. Our ablation studies quantitatively demonstrate that this genomic pre-training is indispensable for achieving high performance. Second, we propose DNABERT-Epi, a multi-modal model that integrates sequence data with epigenetic features, and we rigorously validate that this integration provides a statistically significant improvement in predictive accuracy. Third, recognizing the challenge of comparing models developed under different conditions, we provide a fair and comprehensive benchmark by re-implementing five state-of-the-art models and evaluating them alongside our own under a unified, stringent cross-validation framework [26,27]. Finally, we move beyond simple performance metrics by employing advanced interpretability techniques to provide novel insights into the biological mechanisms learned by the models.

The source codes used in this study are available at https://github.com/kimatakai/CRISPR_DNABERT.

## 2. Materials and methods

### 2.1. Datasets

**2.1.1. Overview of utilized datasets.** In this study, we utilized one *in vitro* and six *in cellula* CRISPR/Cas9 off-target datasets to comprehensively evaluate our proposed models (Table 1). The *in vitro* dataset, derived from CHANGE-seq, was used for the initial training of all models from scratch [23]. For *in cellula* evaluation, we employed a multi-stage approach. First, we used two large-scale datasets, Lazzarotto *et al.* GUIDE-seq and Schmid-Burgk *et al.* TTISS, for training via transfer learning from the CHANGE-seq-trained models [28]. The Lazzarotto *et al.* GUIDE-seq dataset was particularly important as it was used to train and evaluate our multi-modal DNABERT-Epi model, which incorporates epigenetic features. To rigorously assess the generalization performance of our models, we used the remaining four *in cellula* datasets from Chen *et al.*, Listgarten *et al.*, and Tsai *et al.* exclusively as independent test sets [9,29,30].

**2.1.2. Data acquisition and preprocessing.** To ensure a fair and reproducible comparison, we utilized datasets curated by Yaish *et al.* [27]. Specifically, the datasets from Lazzarotto *et al.*, Chen *et al.*, Listgarten *et al.*, and Tsai *et al.* were obtained directly from their repository. For the Tsai *et al.* (2015) GUIDE-seq dataset, we separated the original combined dataset into U2OS and HEK293 cell-specific subsets for more precise evaluation. The Schmid-Burgk *et al.* (2020) TTISS dataset was generated by processing the raw sequence read data from PRJNA602092, following a pipeline identical to that used by Yaish *et al.* The Lazzarotto *et al.* (2020) GUIDE-seq dataset was expanded to include 20 additional sgRNAs newly curated by Yaish *et al.*, resulting in a total of 78 sgRNAs for our 14-fold cross-validation.

All datasets exhibited a significant class imbalance between active (positive) and inactive (negative) off-target sites (Table 1). To mitigate potential model bias during training, we performed random downsampling on the negative class of the training data, reducing its size to 20% of the original. To ensure reproducibility across all models, this downsampling was performed once using a fixed random seed. This strategy to address severe class imbalance is a common approach in various bioinformatics classification tasks [18,19]. The test datasets remained unaltered throughout the process to allow for an unbiased evaluation of model performance.

### 2.2. Epigenetic feature processing

The selection of epigenetic features for our DNABERT-Epi model was guided by the findings of Lazzarotto *et al.* (2020), the source study for our primary *in cellula* dataset. Their research demonstrated that off-target sites identified by GUIDE-seq are significantly enriched in regions characterized by open chromatin (ATAC-seq), active promoters (H3K4me3), and enhancers (H3K27ac) [23]. In contrast, no significant enrichment was observed for repressive histone marks such

**Table 1. Overview of datasets used for training and evaluation.**

| Dataset name | Year | Environment | Cell type | Method | #sgRNAs | #Positive | #Negative |
|---|---|---|---|---|---|---|---|
| **Lazzarotto. CHANGE-seq** | 2020 | *in vitro* | CD4 + /CD8 + T cells | CHANGE-seq | 110 | 202041 | 4936279 |
| **Lazzarotto. GUIDE-seq** | 2020 | *in cellula* | CD4 + /CD8 + T cells | GUIDE-seq | 78 | 2166 | 3271049 |
| **Schmid-Burgk. TTISS** | 2020 | *in cellula* | HEK293T | TTISS | 59 | 1381 | 1518394 |
| **Chen. GUIDE-seq** | 2017 | *in cellula* | U2OS | GUIDE-seq | 6 | 205 | 1741649 |
| **Listgarten. GUIDE-seq** | 2018 | *in cellula* | U2OS | GUIDE-seq | 23 | 86 | 579095 |
| **Tsai. GUIDE-seq (U2OS)** | 2015 | *in cellula* | U2OS | GUIDE-seq | 6 | 265 | 1765441 |
| **Tsai. GUIDE-seq (HEK293)** | 2015 | *in cellula* | HEK293 | GUIDE-seq | 4 | 155 | 170188 |

The table provides a summary of the seven CRISPR/Cas9 off-target datasets utilized in this study. For each dataset, it details the publication year, experimental environment (*in vitro* or *in cellula*), cell type, off-target detection method, and the number of unique single-guide RNAs (sgRNAs), active off-target sites (Positive), and inactive sites (Negative).

as H3K27me3 and H3K9me3. Based on this direct evidence, we focused on integrating these three activating marks to enhance the predictive power of our model. The raw epigenetic data were obtained from the Gene Expression Omnibus (GSE149363).

The processing pipeline for each of the three epigenetic features was as follows. First, for each potential off-target site, we extracted the signal values within a 1000 bp window, centered on the cleavage site (±500 bp). To handle potential outliers within this window, signal values exceeding the range of Q1 - 1.5 * IQR or Q3 + 1.5 * IQR were capped at these respective boundary values. Subsequently, a Z-score transformation was applied to the signal values across the entire dataset for normalization. Finally, the normalized signal within the 1000 bp window was divided into 100 bins of 10 bp each, and the average signal was calculated for each bin, resulting in a 100-dimensional feature vector for each epigenetic mark. These three vectors were then concatenated to form a final 300-dimensional feature vector, which was used as the epigenetic input for the DNABERT-Epi model.

## 2.3. Model architecture

### 2.3.1. DNABERT fine-tuning.
DNABERT is a BERT-based model pre-trained on a large corpus of DNA sequences, enabling it to learn the fundamental patterns of the DNA language [25]. In this study, we utilized the 3-mer DNABERT model, which was pre-trained on a masked language model (MLM) task. To adapt this model for off-target prediction, we implemented a two-stage fine-tuning process (Fig 1A).

The first stage involved fine-tuning the model on a mismatch position prediction task. This task was designed to explicitly teach the model the pairing relationship between sgRNA and target DNA sequences. For this stage, we used a batch size of 8, a learning rate of 2e-5, and trained for five epochs. In the second stage, the model was further fine-tuned for the primary binary classification task of predicting off-target effects. For this off-target prediction task, we used a batch size of 256, a learning rate of 2e-5, and trained for five epochs.

Before fine-tuning, DNABERT's vocabulary was expanded to include 3-mer tokens containing the bulge character ('-') to handle insertions and deletions. As illustrated in Fig 1B, input sequences were formatted by concatenating the 3-mer tokens of the sgRNA and target DNA, separated by special tokens: [CLS] sgRNA 3-mer tokens [SEP] DNA 3-mer tokens [SEP].

### 2.3.2. DNABERT-Epi for multimodal integration.
To investigate the impact of epigenetic context on off-target activity, we propose DNABERT-Epi, a multi-modal model that integrates sequence information with epigenetic features (Fig 1C). This model uses the fine-tuned DNABERT as its sequence-processing backbone.

The architecture of DNABERT-Epi processes two distinct inputs simultaneously. The tokenized sequence is fed into the DNABERT component to generate a high-level sequence representation, from which we extract the final embedding of the [CLS] token. Concurrently, the 300-dimensional epigenetic feature vector is passed through a multi-layer perceptron (MLP) to produce an epigenetic embedding. To control the influence of the epigenetic information based on the sequence context, we employ a gating mechanism. A gate vector is generated from the CLS embedding, which then modulates the epigenetic embedding through element-wise multiplication. Finally, the original CLS embedding and the gated epigenetic embedding are concatenated and fed into a linear layer with a softmax activation function to compute the final probability of an off-target event.

### 2.3.3. Baseline models.
To evaluate the performance of our proposed models, we compared them against five state-of-the-art deep learning-based models for CRISPR off-target prediction: GRU-Embed, CRISPR-BERT, CRISPR-HW, CRISPR-DIPOFF, and CrisprBERT [16,17,27,31,32]. To ensure a fair and direct comparison under identical experimental conditions, we re-implemented all baseline models in PyTorch (version 2.5.1), based on the descriptions in their respective original papers and publicly available source code. As the original implementations of CRISPR-DIPOFF and CrisprBERT did not support inputs containing bulges, we modified their data processing modules accordingly. The hyperparameters used for each baseline model are detailed in Table 2.

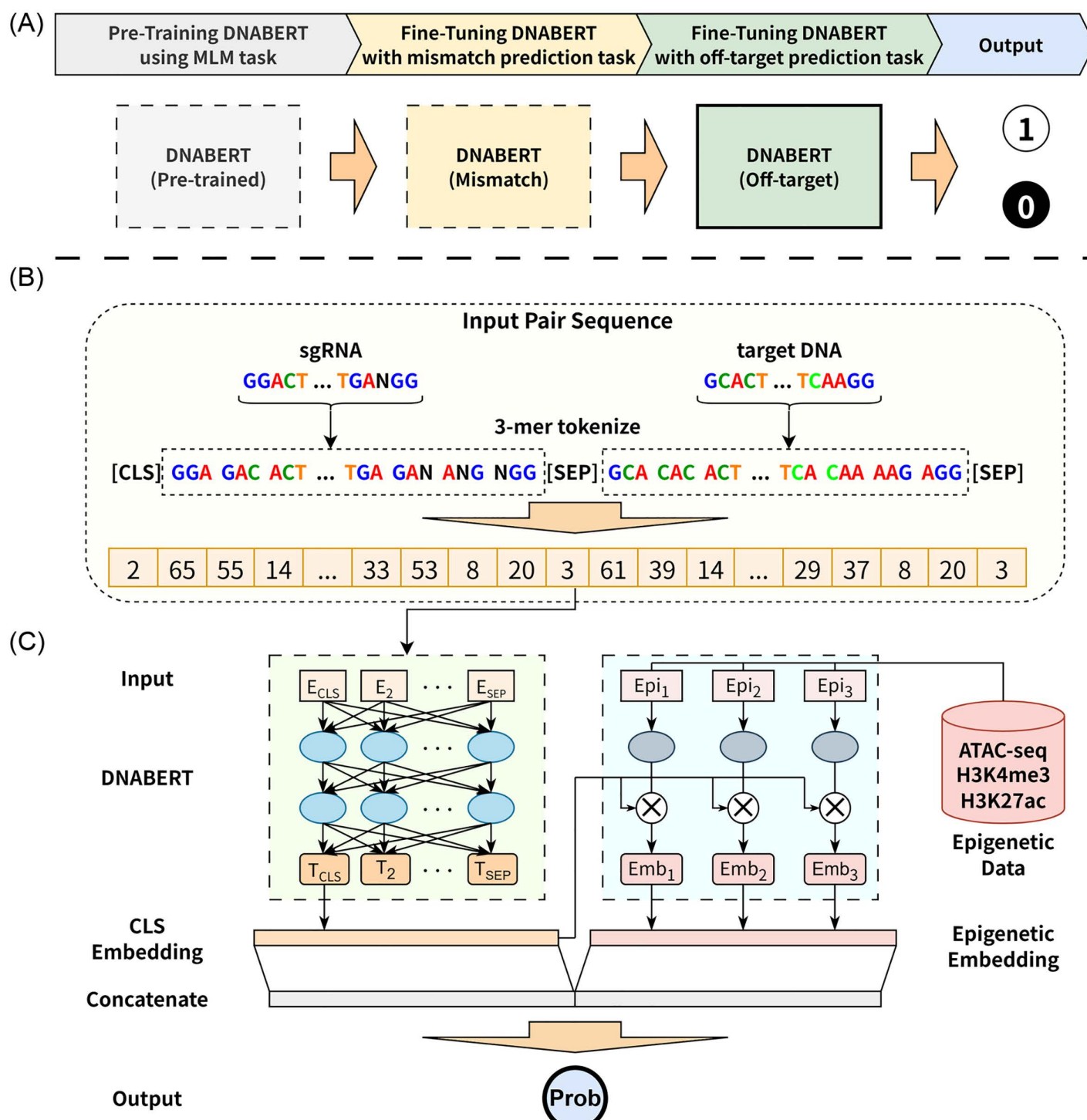

**Fig 1. Overview of the DNABERT Fine-Tuning Process and the DNABERT-Epi Model Architecture. (A)** The two-stage fine-tuning process for DNABERT. A model pre-trained on a masked language model (MLM) task is first fine-tuned on a mismatch position prediction task, followed by a second fine-tuning stage on the off-target effect prediction task to produce a binary output (1 for active, 0 for inactive). **(B)** The input sequence processing pipe-line. The sgRNA and target DNA sequences are first tokenized into 3-mers. These are then formatted with special tokens [CLS] and [SEP] before being converted into numerical input IDs for the model. **(C)** The architecture of the proposed DNABERT-Epi model. The model takes two inputs: the tokenized

sequence, which is processed by DNABERT to produce a CLS embedding, and the epigenetic features, which are processed by a separate MLP. A gating mechanism, derived from the CLS embedding, modulates the epigenetic embedding. Finally, the CLS embedding and the gated epigenetic embedding are concatenated and passed to a final output layer to predict the off-target probability.

**Table 2. Hyperparameters of the baseline models.**

| Model name | Epochs | Batch size | Learning rate |
|---|---|---|---|
| GRU-Embed | 10 | 512 | 0.005 |
| CRISPR-BERT | 30 | 256 | 0.0001 |
| CRISPR-HW | 30 | 128 | 0.003 |
| CRISPR-DIPOFF | 50 | 64 | 0.0001 |
| CrisprBERT | 10 | 128 | 0.00002 |

The table lists the key hyperparameters—training epochs, batch size, and learning rate—used for each of the five baseline models in this study. These settings were primarily adopted from the values reported in the respective original publications and their official source codes. An exception was made for CrisprBERT, for which the number of training epochs was reduced from 400, as specified in the original source code, to 10 to reduce computational cost and prevent overfitting.

**2.3.4. Ensemble model.** To further enhance predictive performance and robustness, we also constructed an ensemble model. We employed a soft voting strategy, where the final prediction is determined by averaging the probability scores output by each individual model. For datasets without epigenetic information, the ensemble combined the five baseline models and DNABERT. For the Lazzarotto *et al*. (2020) GUIDE-seq dataset, the ensemble included all seven models (the five baselines, DNABERT, and DNABERT-Epi). This approach is designed to leverage the complementary strengths of diverse model architectures to achieve a more stable and accurate prediction [33].

## 2.4. Implementation details

All models were implemented using Python 3.10. The deep learning frameworks used were PyTorch (version 2.5.1) and Transformers (version 4.48.3). All training and evaluation experiments were conducted on a workstation equipped with an NVIDIA GeForce RTX 4090 GPU (24 GB of VRAM), running on a Linux operating system with CUDA (version 12.4). The training times for each model on the main datasets are provided in the Supplementary Materials (S4 Table).

## 2.5. Experimental setup and evaluation

**2.5.1. Cross-validation strategy.** To robustly evaluate the predictive performance of the models, we employed a comprehensive, two-tiered validation strategy.

First, for the datasets used in training (Lazzarotto *et al*. (2020) CHANGE-seq, Lazzarotto *et al*. (2020) GUIDE-seq, and Schmid-Burgk *et al*. (2020) TTISS), we implemented a rigorous sgRNA-based cross-validation scheme. This approach ensures that all off-target sites associated with a particular sgRNA are entirely contained within a single fold, preventing information leakage between the training and test sets and thereby providing a more realistic estimate of a model's ability to generalize to new sgRNAs [27]. The number of folds was set to 10 or 14, depending on the dataset, to align with established evaluation protocols.

Second, to assess the true generalization capability of the models on completely unseen data, we performed validation using four independent datasets that were entirely excluded from any training or hyperparameter tuning processes. This comprehensive approach, which combines cross-validation with independent testing, is essential for mitigating the risk of overfitting and ensuring that a model can generalize to new data from different cell types or experimental methods. The importance of such a dual-validation strategy has been emphasized as a critical practice for developing reliable and practical deep learning models in computational biology [18,20].

Furthermore, to account for the stochasticity inherent in deep learning models (e.g., random weight initialization), the entire cross-validation process was repeated five times with different random seeds. The final reported performance metrics represent the aggregated results from these five independent runs.

**2.5.2. Evaluation metrics.** Given the extreme class imbalance inherent in the CRISPR off-target datasets used in this study, we selected four primary evaluation metrics that are well-suited for such scenarios: F1-score, Matthews Correlation Coefficient (MCC), the area under the receiver operating characteristic curve (ROC-AUC), and the area under the precision-recall curve (PR-AUC).

The F1-score, as the harmonic mean of precision and recall, provides a balanced measure of a model's performance when the positive class is rare. The MCC considers all four entries of the confusion matrix (true positives, true negatives, false positives, and false negatives) and is widely regarded as one of the most robust metrics for imbalanced classification. While ROC-AUC assesses the overall discriminative ability of a model across all thresholds, PR-AUC is often more informative in settings with a large skew in the class distribution, as it evaluates the trade-off between precision and recall. By utilizing this suite of metrics, we ensure a comprehensive and reliable assessment of model performance.

**2.5.3. Statistical analysis.** To rigorously compare the performance between different models, we employed statistical tests appropriate for the paired nature of our cross-validation results (i.e., all models were evaluated on the same data folds). A two-sided Wilcoxon signed-rank test was used to determine if there were statistically significant differences between the performance distributions of any two models.

Furthermore, to address the issue of multiple comparisons arising from performing tests across numerous model pairs and metrics, we controlled the false discovery rate (FDR) using the Benjamini-Hochberg (BH) procedure. All reported p-values are adjusted p-values from this procedure, ensuring the statistical robustness of our conclusions. A significance level of $p < 0.05$ was used throughout the study.

## 2.6. Model interpretability analysis

To gain insights into the decision-making processes of our models beyond predictive accuracy, we employed two distinct interpretability methods. We used SHAP to quantify the contribution of epigenetic features and Integrated Gradients to attribute the prediction to specific sequence tokens.

**2.6.1. SHAP analysis for epigenetic features.** Shapley Additive Explanations (SHAP) is a game theory-based approach used to explain the output of any machine learning model by assigning an importance value to each feature for a particular prediction [34]. To analyze the influence of epigenetic features on the predictions of our DNABERT-Epi model, we utilized the DeepExplainer algorithm, which is an efficient variant of SHAP designed for deep learning models [35].

For the analysis, we first extracted a balanced subset of data, consisting of all active off-target sites and an equal number of randomly sampled inactive sites from the Lazzarotto *et al*. GUIDE-seq dataset. We then calculated the SHAP values for the 300-dimensional epigenetic input features for each sample in this subset. To derive global insights, these SHAP values were aggregated in two ways: (1) the mean absolute SHAP values were calculated for each of the three epigenetic marks (ATAC-seq, H3K4me3, and H3K27ac) to determine their overall importance, and (2) the mean SHAP values for each of the 100 genomic bins were computed to visualize the positional importance of epigenetic signals relative to the cleavage site.

**2.6.2. Integrated gradients for sequence features.** Integrated Gradients (IG) is a feature attribution method that calculates the importance of each input feature by accumulating the gradients along the path from a baseline input to the actual input [36]. We applied IG to the DNABERT model to identify which nucleotide tokens were most influential in predicting active off-target sites.

The embedding of the [PAD] token was used as the baseline for this analysis. For each active off-target site, we calculated the attribution score for every 3-mer token in the input sequence. These scores were then visualized as heatmaps to reveal attribution patterns for individual sgRNAs. To statistically validate the significance of recurrently high-attribution

regions (hotspots), we conducted a randomization test with 1,000 iterations. In each iteration, we compared the mean attribution in these regions to randomly selected regions of the same size to calculate an empirical p-value. Finally, to explore overarching patterns across all sgRNAs, the attribution vectors were embedded into a two-dimensional space using Uniform Manifold Approximation and Projection (UMAP) for visualization and clustering [37].

## 3. Results

### 3.1. Benchmarking performance on diverse datasets

To establish the effectiveness of our proposed methods, we conducted a comprehensive benchmark against five state-of-the-art models across seven distinct datasets. First, we focused on the Lazzarotto *et al*. GUIDE-seq dataset to evaluate our multi-modal DNABERT-Epi model. As shown in Fig 2, DNABERT-Epi demonstrated superior performance, particularly in metrics sensitive to class imbalance. It achieved a significantly higher F1-score and MCC compared to the sequence-only DNABERT model ($p < 0.05$), and its PR-AUC of 0.550 was the highest among all single models, outperforming DNA-BERT (0.539) and CrisprBERT (0.511). While several models, including DNABERT-Epi, achieved high ROC-AUC scores, PR-AUC provides a more critical evaluation in this context. The ensemble model consistently achieved the best performance across all metrics, highlighting the benefits of integrating diverse model predictions. Detailed performance metrics and the complete results of statistical tests for this dataset are provided in S1 Table and S2 Table, respectively.

Next, to assess the generalization performance on the remaining datasets, we evaluated the sequence-based models on the *in vitro* CHANGE-seq data and the five other *in cellula* datasets. The overall trend, summarized by the PR-AUC in Fig 3, indicates that DNABERT is a consistently strong performer across these diverse conditions. For instance, on the Schmid-Burgk 2020 TTISS and Tsai 2015 GUIDE-seq (HEK293) datasets, DNABERT achieved the highest PR-AUC among all single models. However, the performance varied depending on the dataset; CrisprBERT, for example, showed competitive performance on the Chen 2017 and Tsai 2015 U2OS GUIDE-seq datasets. This highlights that no single model architecture is universally optimal for all conditions. Importantly, our proposed DNABERT model maintained robust and competitive performance across this wide range of datasets derived from different experimental methods and cell types, confirming its high generalization capability. The comprehensive performance results for all models across all metrics and datasets are available in the Supplementary Materials, which include detailed performance plots (S1 File), full numerical results (S1 Table), statistical test p-values (S2 Table), and confusion matrices organized by mismatch count (S3 Table).

### 3.2. Ablation studies confirm key contributions

To dissect the factors contributing to our model's performance, we conducted two key ablation studies. These studies were designed to quantitatively assess the impact of DNABERT's pre-training and the integration of epigenetic features.

First, to evaluate the effectiveness of leveraging a pre-trained foundation model, we compared the performance of our fine-tuned DNABERT model against an identical model architecture trained from scratch (i.e., with randomly initialized weights). The results, summarized in Table 3, demonstrate the critical importance of pre-training. The model initialized with pre-trained weights substantially outperformed the from-scratch model across all evaluation metrics (e.g., +0.1653 in PR-AUC; $p < 0.001$). While the from-scratch model showed evidence of some learning, its performance was markedly inferior to the pre-trained model. This suggests that the genomic knowledge encoded during pre-training is essential for the model to effectively learn the complex and imbalanced off-target prediction task, a conclusion supported by the less effective training loss reduction observed for the from-scratch model (S1 Fig).

Second, we conducted an ablation study to determine whether the inclusion of epigenetic information provides a tangible benefit for off-target prediction in a cellular context. We compared the performance of the sequence-only DNABERT model with our multi-modal DNABERT-Epi model on the Lazzarotto *et al*. (2020) GUIDE-seq dataset. As shown in Table 4, the integration of epigenetic features led to statistically significant improvements in three of the four key metrics: F1-score, MCC, and PR-AUC ($p < 0.001$). No statistically significant difference was observed in ROC-AUC ($p = 0.127$). These results

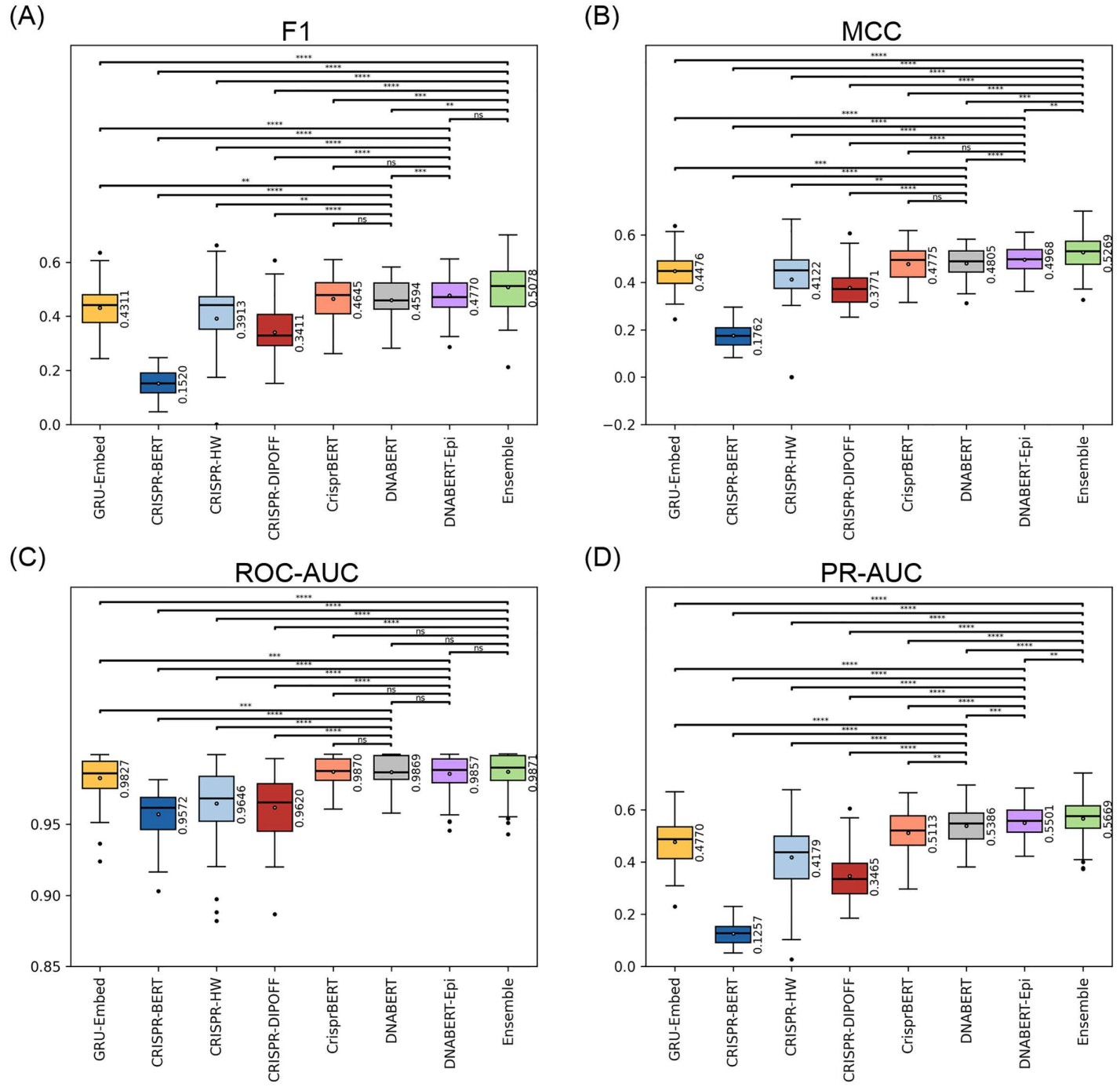

**Fig 2. Performance comparison of all models on the Lazzarotto *et al.* (2020) GUIDE-seq dataset.** Boxplots show the distribution of **(A)** F1-score, **(B)** MCC, **(C)** ROC-AUC, and **(D)** PR-AUC scores from the 14-fold cross-validation experiments. The central line in each box indicates the median, the box represents the interquartile range (IQR), and the whiskers extend to 1.5 times the IQR. Dots beyond the whiskers are outliers. Statistical significance between model pairs was determined using the two-sided Wilcoxon signed-rank test with Benjamini-Hochberg correction. Significance levels are denoted as follows: ns: p > 0.05, *: p ≤ 0.05, **: p ≤ 0.01, ***: p ≤ 0.001, ****: p ≤ 0.0001.

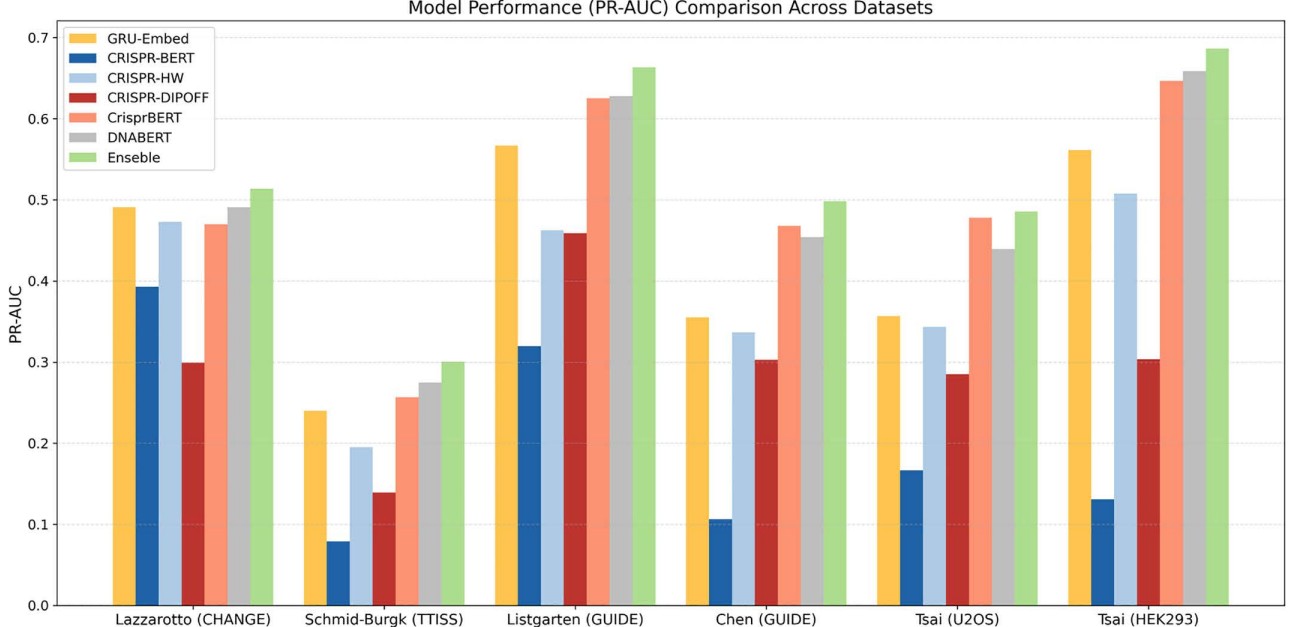

**Fig 3. Comparison of PR-AUC performance on training and independent test datasets.** This grouped bar chart provides a summary of model performance using the Precision-Recall Area Under the Curve (PR-AUC). The figure displays results for six datasets, facilitating a direct comparison of the models' generalization capabilities. The Lazzarotto *et al.* (2020) GUIDE-seq dataset, which was analyzed in detail in Fig 2, is excluded from this summary view. Each bar represents the mean PR-AUC score for a given model on a specific dataset, calculated from the results of the cross-validation experiments. Error bars indicate the standard deviation of the PR-AUC scores across the folds. Higher bars indicate superior predictive performance. The labels on the x-axis are abbreviated for space and correspond to Lazzarotto *et al.* (2020) CHANGE-seq, Schmid-Burgk *et al.* (2020) TTISS, List-garten *et al.* (2018) GUIDE-seq, Chen *et al.* (2017) GUIDE-seq, Tsai *et al.* (2015) GUIDE-seq (U2OS), and Tsai *et al.* (2015) GUIDE-seq (HEK293), respectively.

**Table 3. Performance comparison of DNABERT with and without pre-training.**

|  | F1 | MCC | ROC-AUC | PR-AUC |
|---|---|---|---|---|
| **DNABERT (Pre-trained)** | $0.4516 \pm 0.1155$ | $0.4478 \pm 0.1037$ | $0.9234 \pm 0.0226$ | $0.4911 \pm 0.1421$ |
| **DNABERT (from scratch)** | $0.3546 \pm 0.1287$ | $0.3412 \pm 0.1192$ | $0.8753 \pm 0.0315$ | $0.3257 \pm 0.1480$ |
| **Difference** | $+0.09697$ | $+0.1066$ | $+0.0481$ | $+0.1653$ |
| **P-Value** | $1.9607 \times 10^{-12}$ | $4.4187 \times 10^{-13}$ | $3.5595 \times 10^{-13}$ | $3.5595 \times 10^{-13}$ |

The table shows the performance of the DNABERT model on the off-target prediction task when initialized with pre-trained weights versus random weights (from scratch). Both models were fine-tuned and evaluated on the Lazzarotto *et al.* (2020) CHANGE-seq dataset. Performance is reported as mean ± standard deviation, along with the absolute difference and the p-value from a Wilcoxon signed-rank test.

confirm that incorporating epigenetic context enhances the model's ability to distinguish between active and inactive off-target sites *in cellula*, particularly in ways captured by precision-recall-based metrics.

### 3.3. Ensemble model benefits from diverse architectures

Our benchmark results consistently showed that the ensemble model outperformed any single model across all datasets (Fig 2, Fig 3). To understand the dynamics within the ensemble, we analyzed the contribution of each constituent model using a leave-one-out strategy. This analysis reveals how the diversity of model architectures contributes to the overall superior performance of the ensemble, a phenomenon that has been noted in previous studies [33].

**Table 4. Performance comparison of DNABERT with and without epigenetic features.**

|  | F1 | MCC | ROC-AUC | PR-AUC |
|---|---|---|---|---|
| **DNABERT-Epi** | 0.4770±0.0697 | 0.4968±0.0586 | 0.9857±0.0124 | 0.5501±0.0673 |
| **DNABERT** | 0.4594±0.0763 | 0.4805±0.0639 | 0.9869±0.0104 | 0.5386±0.0684 |
| **Difference** | +0.0176 | +0.0163 | −0.0012 | +0.0115 |
| **P-Value** | $1.0963 \times 10^{-4}$ | $5.4124 \times 10^{-5}$ | $1.2703 \times 10^{-1}$ | $7.9846 \times 10^{-4}$ |

This table presents an ablation study evaluating the contribution of epigenetic features by comparing the performance of the sequence-only DNABERT model with the multi-modal DNABERT-Epi model. Both models were evaluated on the Lazzarotto *et al.* (2020) GUIDE-seq dataset. Performance is reported as mean±standard deviation, along with the absolute difference and the p-value from a Wilcoxon signed-rank test.

Fig 4 illustrates the contribution of each model to the ensemble's performance on the Lazzarotto *et al.* (2020) GUIDE-seq dataset, as measured by the drop in PR-AUC when a model is excluded. As expected, high-performing individual models such as CrisprBERT and our proposed DNABERT-Epi were the largest contributors. Their exclusion led to the most significant drop in the ensemble's performance, confirming their central role. Notably, the analysis also revealed that models with moderate individual performance, such as GRU-Embed and CRISPR-HW, still made substantial positive contributions. This suggests that these models capture unique predictive patterns that are complementary to those learned by the top-performing models. Conversely, the exclusion of CRISPR-BERT slightly improved the ensemble's score, indicating that its predictions were, on average, detrimental in this specific combination. This analysis underscores the principle that the strength of an ensemble lies not just in combining the best-performing models, but in leveraging the diverse perspectives of multiple architectures. The detailed results of this analysis for all datasets are provided in the S2 File.

### 3.4. Model interpretation unveils predictive mechanisms

To move beyond predictive accuracy and understand the biological patterns learned by our models, we employed two interpretability techniques. We used SHAP to analyze the role of epigenetic features in DNABERT-Epi and Integrated Gradients to identify critical nucleotide positions in the sequence-only DNABERT model.

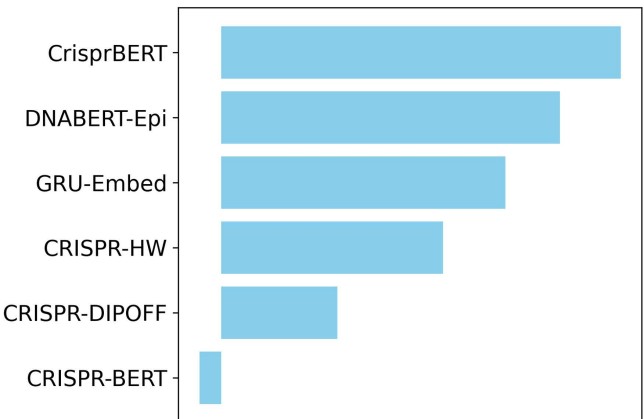

**Fig 4. Contribution of individual models to the ensemble performance on the Lazzarotto *et al*. (2020) GUIDE-seq dataset.** The contribution of each model to the final ensemble was assessed using a leave-one-out approach. Each bar represents the decrease in the ensemble's PR-AUC score when that specific model is excluded from the soft-voting process. A longer bar indicates a greater positive contribution to the ensemble's predictive power. Models are ranked by their contribution.

**3.4.1. SHAP reveals the critical role of H3K27ac.** To elucidate how DNABERT-Epi utilizes epigenetic information, we calculated SHAP values for each input feature. This approach is similar to previous studies in RNA modification prediction, where SHAP has been used to link computational predictions to biologically meaningful motifs [20]. The analysis of global feature importance revealed a clear hierarchy among the three epigenetic marks. As shown in Fig 5A, H3K27ac, a mark associated with active enhancers, had a substantially higher mean absolute SHAP value than H3K4me3 (active promoters) and ATAC-seq (open chromatin), identifying that H3K27ac is the most influential feature in the model. This is further supported by the SHAP summary plot (Fig 5B and S3 File), where the top 30 most impactful features consist exclusively of H3K27ac-related bins. The plot also shows a consistent trend where higher signal values for these features (red points) positively impact the model's prediction of off-target activity.

Examining the positional importance of these features (Fig 5C), H3K27ac consistently showed the highest mean SHAP values across the entire ±500 bp window around the cleavage site. Importantly, the contribution of H3K27ac was not uniform, with prominent peaks observed around ±200 bp and near the ±500 bp boundaries of the window. These

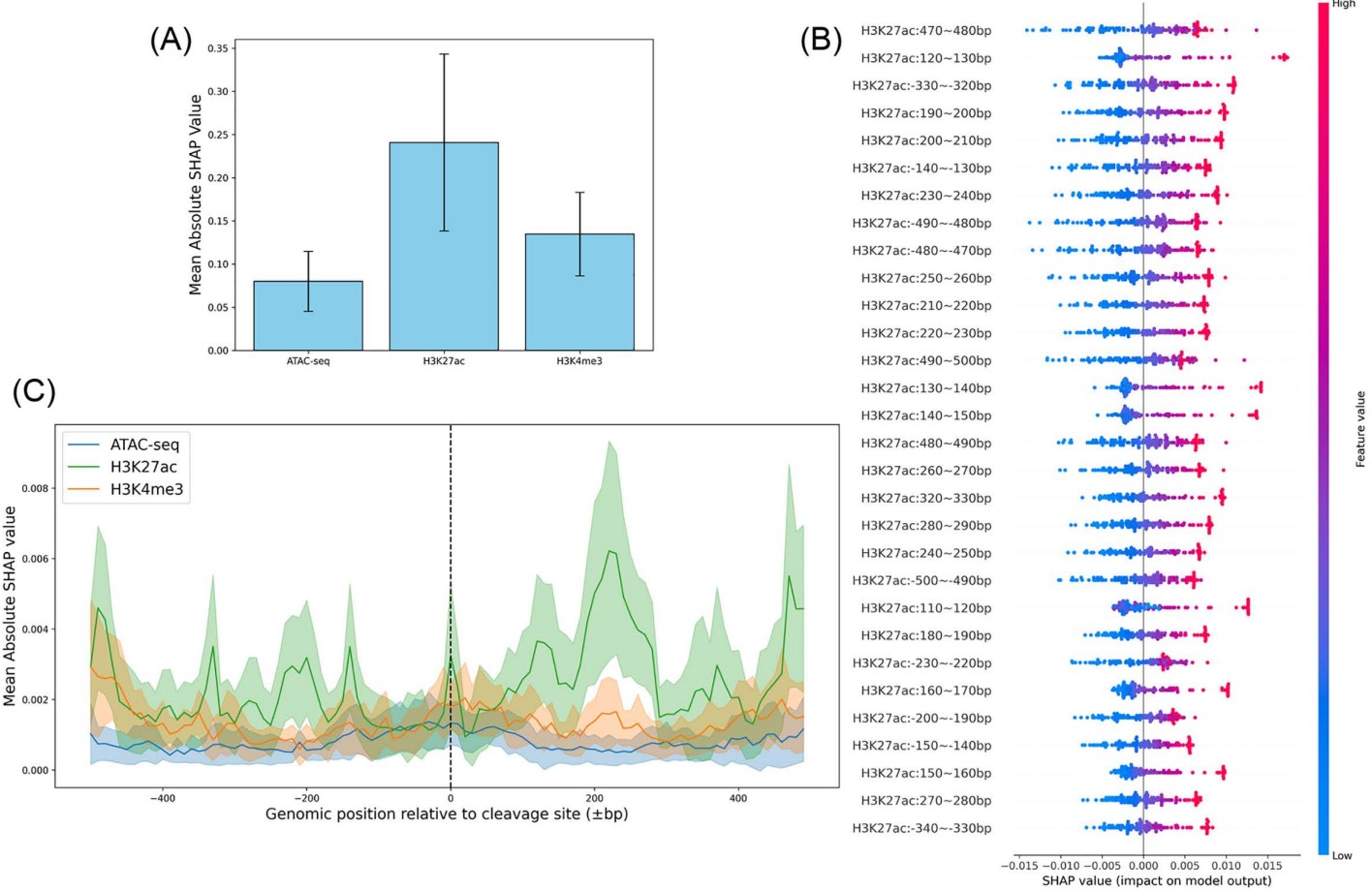

**Fig 5. SHAP analysis of epigenetic feature contributions in the DNABERT-Epi model.** The analysis was performed on the Lazzarotto *et al*. (2020) GUIDE-seq dataset. **(A)** The global importance of each epigenetic mark, measured by the mean absolute SHAP value across all features and samples. Error bars represent the standard deviation. **(B)** A SHAP summary plot from a representative cross-validation fold, illustrating the impact of the top 30 most important feature bins. Each point is a single sample, with its color indicating the feature's value (red for high, blue for low) and its x-position showing the impact on the model's output. **(C)** The positional importance of each epigenetic mark. The plot shows the mean absolute SHAP value for each 10 bp bin across a ±500 bp window centered on the cleavage site. The shaded area represents the 95% confidence interval.

results suggest that the model has learned to associate off-target events not only with the general presence of an active enhancer landscape but also with specific spatial patterns of H3K27ac enrichment relative to the cleavage site.

**3.4.2. Integrated gradients identify key nucleotide positions and sgRNA clusters.** To understand which parts of the input sequence were most critical for prediction, we used Integrated Gradients to calculate attribution scores for each 3-mer token. The resulting heatmaps for individual sgRNAs consistently revealed two distinct high-attribution regions, or "hotspots," within the 20-nt guide-target duplex (Fig 6A, 6B and S4 File). The first hotspot was located at the PAM-distal end (positions 4–6), while the second was at the PAM-proximal end (positions 14–17), partially overlapping with the canonical seed region. A randomization test confirmed that the attribution scores within these two hotspots were, in most cases (15 of 18 data subsets), statistically significantly higher than in the rest of the sequence (all p-values in S5 Table).

To investigate whether these attribution patterns were consistent across all sgRNAs, we visualized the high-dimensional attribution vectors using UMAP (Fig 6C). The plot reveals that the sgRNAs form three distinct clusters: one group where the PAM-distal hotspot has the maximum attribution (red), another where the PAM-proximal hotspot is dominant (blue), and a third with no clear hotspot pattern (grey). This clustering suggests that the model does not apply a single, uniform pattern of sequence recognition across all sgRNAs. Instead, the model appears to have learned to weigh nucleotide importance in a context-dependent manner. This finding may indicate that the sequence determinants of off-target activity are more nuanced than a simple binary seed/non-seed distinction, and that the model is capturing elements of this complexity.

## 4. Discussion

This study presents a comprehensive evaluation of a pre-trained DNA foundation model, DNABERT, for CRISPR/Cas9 off-target prediction. We have demonstrated that by leveraging large-scale genomic pre-training and integrating epigenetic features, our proposed models, DNABERT and DNABERT-Epi, achieve state-of-the-art performance across a wide range of datasets. Our contributions can be summarized in fourfold: (1) we are the first to apply a pre-trained DNA foundation model to this task; (2) we have quantitatively demonstrated the significant contribution of both pre-training and epigenetic features through rigorous ablation studies; (3) we have provided a fair and extensive benchmark by re-implementing and evaluating multiple models under identical conditions; and (4) we have offered novel insights into the models' decision-making processes through advanced interpretability analyses.

Our interpretability analyses provided valuable insights into the predictive mechanisms of the models. The SHAP analysis of DNABERT-Epi revealed that H3K27ac, an active enhancer mark, emerged as the most influential feature among those examined for predicting off-target activity in a cellular context. This aligns with existing biological knowledge that CRISPR/Cas9 activity is often higher in accessible, transcriptionally active chromatin regions [23,38,39]. Furthermore, our analysis of positional importance suggested that the model learned specific spatial patterns of H3K27ac enrichment, rather than just its overall presence.

Regarding sequence specificity, the canonical "seed" region adjacent to the PAM is widely considered the most critical determinant for target recognition [7,40–47]. We therefore initially hypothesized that our interpretability analysis would primarily and uniformly highlight this region. However, the Integrated Gradients analysis of DNABERT revealed a more nuanced picture. Rather than a singular focus on the entire seed region, the model identified two distinct attribution hotspots: a PAM-proximal (positions 14–17) and a PAM-distal (positions 4–6) region. These computationally identified hotspots may reflect critical stages in R-loop formation and conformational activation, although this remains a hypothesis requiring further experimental validation [42,48–50]. For instance, the high importance placed on the PAM-proximal hotspot is consistent with recent structural studies that describe a conformational checkpoint mechanism, where mismatches in this region can prevent the activation of the HNH nuclease domain [48–50]. This consistency suggests, but does not prove, that the model may be capturing sequence features relevant to this checkpoint. In parallel, the PAM-distal hotspot could represent the model's focus on an earlier stage, such as the initiation and stable propagation of the R-loop.

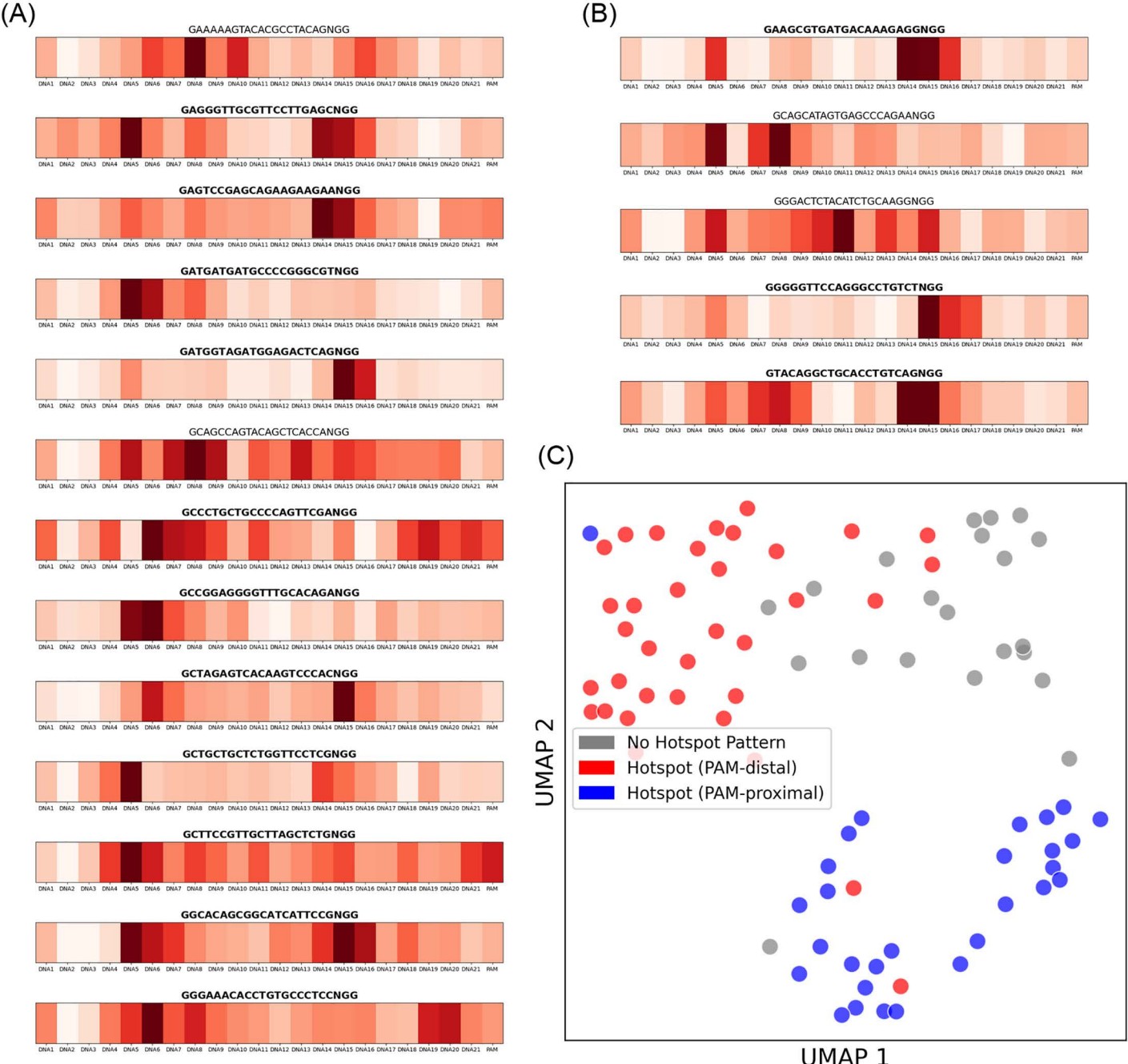

**Fig 6. Integrated Gradients attribution analysis of sequence features.** **(A)** and **(B)** show representative attribution heatmaps for individual sgRNAs from the Lazzarotto *et al*. (2020) GUIDE-seq and Listgarten *et al*. (2018) GUIDE-seq datasets, respectively. The columns correspond to the 3-mer token positions in the target DNA. Color intensity reflects the attribution score, with darker red indicating a stronger positive contribution to the prediction of an off-target event. The sgRNA sequence is shown above each map. **(C)** UMAP visualization of the attribution vectors from all analyzed sgRNAs. Each point represents one sgRNA. Points are colored based on the location of their maximum attribution score: red for the PAM-distal hotspot (positions 4–6), blue for the PAM-proximal hotspot (positions 14–17), and grey for all others.

Interestingly, the UMAP visualization of attribution vectors further suggested that sgRNAs may cluster into three groups: those emphasizing the PAM-proximal hotspot, those emphasizing the PAM-distal hotspot, and those without a clear hotspot preference. This preliminary observation may point to a potential classification of sgRNAs into context-dependent categories, although further experimental evidence will be required to establish such a framework. This convergence of our model's learned patterns with biophysical mechanisms may provide a data-driven hypothesis for the nuanced rules governing Cas9 target engagement, but further experimental validation will be necessary to confirm this link.

While DNABERT and DNABERT-Epi demonstrated robust performance, they did not universally outperform all other models across metrics and datasets. For instance, CrisprBERT, a model that does not use large-scale pre-training but combines a BERT-based encoder with a BiLSTM, showed competitive or superior performance on certain datasets. This suggests that while pre-training provides a powerful, generalizable foundation, task-specific architectural innovations are also crucial. A promising future direction would be to combine the strengths of both approaches, for example, by integrating a BiLSTM layer into the fine-tuned DNABERT architecture to better capture sequential dependencies specific to off-target recognition. Our study has several limitations that represent important areas for future research. First, our analysis was focused exclusively on the Streptococcus pyogenes Cas9 (SpCas9) nuclease. The applicability of our models to other CRISPR systems, such as Cas12a or high-fidelity Cas9 variants, remains unverified and will require retraining and evaluation on variant-specific datasets [51–55]. Second, the validation of our multi-modal DNABERT-Epi model was limited to a single GUIDE-seq dataset due to the lack of publicly available, matched epigenetic data for other off-target datasets. The full potential and generalizability of this approach can only be realized as more comprehensive, multi-modal datasets become available. A promising future direction to overcome this data scarcity would be to leverage large-scale, sequence-based predictive models, such as DeepSEA, to generate *in silico* chromatin profiles for these datasets. Integrating these high-quality predicted features could significantly expand the applicability of our multi-modal approach [56]. Third, the high performance of DNABERT-based models comes at a significant computational cost, requiring substantially more training time than other baseline models (S4 Table). Applying model compression techniques such as distillation or pruning could be a valuable next step to create more efficient yet powerful models for broader use [57,58]. Finally, like all models benchmarked in this study, our approach struggled to accurately predict active off-target sites with a high number of mismatches (5–6), frequently misclassifying them as inactive (S3 Table). This "high-mismatch problem" is a key challenge for the field and may require novel architectural solutions or specialized sampling strategies to address effectively.

In conclusion, this work establishes the significant potential of pre-trained foundation models for advancing CRISPR/Cas9 off-target prediction. We have demonstrated that combining the vast genomic knowledge learned during pre-training with context-specific epigenetic information leads to a more accurate and robust prediction of off-target events. The interpretability analyses not only increase transparency but also generate new and testable hypotheses about the underlying biology. Future efforts should focus on expanding these models to other CRISPR variants, improving computational efficiency, and developing new strategies to tackle the persistent challenge of high-mismatch prediction, ultimately contributing to the development of safer and more effective genome editing therapies. A promising direction is to extend our model to predict not just the presence of off-target effects, but also their strength. This approach, which involves a multi-level classification task, has been successfully applied in other domains, such as the identification of enhancers, where a two-stage framework was used to classify both the presence and strength of enhancers [21].

## Supporting information

**S1 File. Performance comparison across all datasets.**
(PDF)

**S2 File. Ensemble contribution (PR-AUC) across all datasets.**
(PDF)

**S3 File.  SHAP summary plots for all datasets.**
(PDF)

**S4 File.  Integrated Gradients heatmaps for all sgRNAs.**
(PDF)

**S1 Table.  Comprehensive performance comparison table across all datasets.**
(XLSX)

**S2 Table.  P-values from statistical comparisons across all datasets.**
(XLSX)

**S3 Table.  Confusion matrices by mismatch count for all datasets.**
(XLSX)

**S4 Table.  Training time comparison for each model.**
(XLSX)

**S5 Table.  P-values from the randomization test for Integrated Gradients analysis.**
(XLSX)

**S1 Fig.  Training loss curves comparing the pre-trained and from-scratch DNABERT models.**
(TIFF)

## Acknowledgments

In this research, the super-computing resource was provided by Human Genome Center, the Institute of Medical Science, the University of Tokyo. In addition, computations were partially performed on the NIG supercomputer at ROIS National Institute of Genetics.

## Author contributions

**Conceptualization:** Kai Kimata.

**Data curation:** Kai Kimata.

**Formal analysis:** Kai Kimata.

**Investigation:** Kai Kimata.

**Methodology:** Kai Kimata.

**Project administration:** Kai Kimata.

**Software:** Kai Kimata.

**Supervision:** Kenji Satou.

**Validation:** Kai Kimata.

**Visualization:** Kai Kimata.

**Writing – original draft:** Kai Kimata.

**Writing – review & editing:** Kai Kimata.

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
