## [Decision Letter · Decision Letter 0]

19 Aug 2025

Dear Dr. Satou,

Thank you for submitting your manuscript to PLOS ONE. After careful consideration, we feel that it has merit but does not fully meet PLOS ONE’s publication criteria as it currently stands. Therefore, we invite you to submit a revised version of the manuscript that addresses the points raised during the review process.

Your manuscript was reviewed by two experts. While they found it potentially interesting, they raised some concerns. Please revise it according to their suggestions.

We look forward to receiving your revised manuscript.

Kind regards,

Hodaka Fujii, M.D., Ph.D.

Academic Editor

PLOS ONE

Journal Requirements: 

Reviewers' comments:

Reviewer's Responses to Questions

**Comments to the Author**

1. Is the manuscript technically sound, and do the data support the conclusions?

Reviewer #1: Yes

Reviewer #2: Yes

2. Has the statistical analysis been performed appropriately and rigorously?

Reviewer #1: No

Reviewer #2: Yes

3. Have the authors made all data underlying the findings in their manuscript fully available?

Reviewer #1: Yes

Reviewer #2: Yes

4. Is the manuscript presented in an intelligible fashion and written in standard English?

Reviewer #1: No

Reviewer #2: Yes

Reviewer #1: The author should focus on making the abstract more concise and targeted, highlighting the main points and emphasizing the major contributions of the proposed model.

2. The manuscript does not address potential limitations or areas for improvement in the proposed system, which should be discussed to provide a balanced perspective.

3. Details on the computational complexity and reliance on specific datasets during data processing or benchmarking need to be included.

4. Relevant literature such as s13040-024-00415-8, s12859-024-05917-0, and s12859-024-05978-1 using bioinformatics datasets for classification problems should be reviewed and compared in the literature section.

5. The integration of DNABERT with epigenetic features is promising, but the novelty is limited as similar transformer-based approaches have been recently explored in CRISPR off-target prediction.

6. The manuscript would benefit from more detailed comparisons with current literature, including deeper analysis of why the proposed method outperforms existing models.

7. While attention weight visualization is a good step toward interpretability, the biological relevance of the patterns identified needs further validation or expert annotation.

8. The writing is generally clear, but some sections—especially the methodology—could be more concise and better organized with clearer subheadings.

Reviewer #2: This manuscript presents a DNABERT-based model that incorporates epigenetic features to improve CRISPR/Cas9 off-target prediction. The proposed approach is timely and technically well-motivated. The following points should be addressed to further strengthen the work:

1.While the use of DNABERT and epigenetic features is compelling, the authors could consider more clearly articulating how their approach differs from previous BERT-based off-target prediction models in terms of integration strategy or architecture.

2.The rationale for choosing only H3K4me3, ATAC-seq, and H3K27ac is underexplored. Please explain why these three features were selected over others and discuss their individual contributions to performance.

3.The model was tested primarily on CHANGE-seq and GUIDE-seq datasets. Would the approach generalize to other datasets or Cas variants? A brief discussion on applicability beyond Cas9 would strengthen the impact.

4.The attention weight visualization is promising but limited to seed/non-seed distinctions. Including more quantitative or alternative interpretation techniques would enhance model transparency.

5.The ensemble method consistently outperforms individual models, but the contribution of each component remains unclear. An ablation of the ensemble composition would better justify its use.

6.The authors are also encouraged to cite recent advances in sequence-based RNA/DNA function or modification prediction, such as https://doi.org/10.1021/acs.jcim.4c00546 and https://doi.org/10.1016/j.csbj.2025.02.029.

**Do you want your identity to be public for this peer review?** For information about this choice, including consent withdrawal, please see our Privacy Policy

Reviewer #1: No

Reviewer #2: No

---

## [Author Response · Author response to Decision Letter 1]

6 Oct 2025

Dear Editor and Reviewers,

Thank you for your insightful and detailed feedback on our manuscript, "Improved CRISPR/Cas9 Off-target Prediction with DNABERT and Epigenetic Features" (Manuscript ID: PONE-D-25-35489). The comments provided by the reviewers were invaluable and have significantly helped us to improve the quality of our work.

Based on your suggestions, we have undertaken a major revision of the manuscript. Key revisions include a complete rewriting of the Abstract, a rigorous validation of generalization performance by adding new datasets, the introduction of advanced interpretability analyses (SHAP and Integrated Gradients) with deeper biological discussion, and a clarification of the overall structure and writing.

Herein, we provide a point-by-point response to each of the reviewers' comments, detailing the corresponding revisions made to the manuscript.

Responses to Reviewer #1

Comment 1: The author should focus on making the abstract more concise and targeted, highlighting the main points and emphasizing the major contributions of the proposed model.

Response 1: Thank you for this valuable suggestion. We agree that the original abstract could have been more focused and impactful. In response to your comment, we have completely rewritten the abstract to be more concise and to clearly emphasize the primary contributions of our work.

Specifically, the revised abstract is now structured to first present the problem, then our novel approach, followed by the key findings, and finally the overall impact. We have explicitly highlighted our three major contributions: (1) the first application of a pre-trained DNA foundation model to this task, (2) the quantitative confirmation of the importance of both pre-training and epigenetic features through ablation studies, and (3) the generation of biological insights via advanced interpretability analyses. We believe the revised abstract now provides a much clearer and more compelling summary of our study.

Comment 2: The manuscript does not address potential limitations or areas for improvement in the proposed system, which should be discussed to provide a balanced perspective.

Response 2: This is an important point, and we thank you for raising it. To provide a more balanced perspective and acknowledge the scientific rigor, we have added a dedicated paragraph in the Discussion section to address the limitations of our study and suggest areas for future work (referencing p.28 lines 494-511). We now discuss four key limitations: (1) the focus on SpCas9, excluding other variants; (2) the restriction of our multi-modal validation to a single dataset; (3) the computational cost of our model (referencing Table S5); and (4) the challenge of predicting sites with a high number of mismatches (referencing Table S4). We also propose potential solutions, such as leveraging predictive models like DeepSEA and extending our work to predict the strength of off-target effects

Comments 3: Details on the computational complexity and reliance on specific datasets during data processing or benchmarking need to be included.

Response 3: Thank you for this suggestion. We have added details regarding computational complexity and dataset reliance. To address computational complexity, we have taken a multi-faceted approach. First, for transparency, we have introduced a new section, “2.4. Implementation Details” (referencing p.11 lines 188-193), which specifies the hardware (NVIDIA GeForce RTX 4090) and software versions used. Second, we provide the practical training times in the Supplementary Materials (Table S5) and have clarified that the training time scales linearly with the number of input samples (O(n)), which ensures the scalability of our approach to larger datasets. The high computational cost of DNABERT-based models is also discussed as a limitation in the Discussion (referencing p.29 lines 504-507). Regarding dataset reliance, we now explicitly state in the Discussion that our analysis using epigenetic features was reliant on a specific dataset due to data availability (referencing p.28 lines 497-504).

Comment 4: Relevant literature such as s13040-024-00415-8, s12859-024-05917-0, and s12859-024-05978-1 using bioinformatics datasets for classification problems should be reviewed and compared in the literature section.

Response 4: We are grateful for these literature suggestions. The three recommended papers are indeed highly relevant to our work. We have reviewed them and incorporated them into our manuscript to support and contextualize our methodological choices. Specifically, these papers reinforce the importance of established practices for handling imbalanced bioinformatics datasets, such as the downsampling strategy we employed (referencing p.5 lines 87-89). Furthermore, they underscore the critical need for validation on independent test sets to ensure model generalization, which was a core component of our revised evaluation framework (referencing p.12 lines 209-211). We have cited these papers in the Introduction (referencing p.3 lines 39-41) and Methods sections where these topics are discussed, providing a broader context for our deep learning-based classification approach.

Comment 5: The integration of DNABERT with epigenetic features is promising, but the novelty is limited as similar transformer-based approaches have been recently explored in CRISPR off-target prediction.

Response 5: We acknowledge that the novelty of our work needed to be articulated more clearly. We have revised the Introduction (referencing p.5 lines 48-52) to emphasize three key points that distinguish our study: (1) This is the first application of a pre-trained DNA foundation model (trained on the entire human genome) to this task, as opposed to a Transformer trained from scratch on task-specific data. (2) We provide the first quantitative validation of the contributions of both pre-training and epigenetic feature integration through rigorous ablation studies. (3) We offer a fair and comprehensive benchmark by re-implementing all baseline models under identical conditions.

Comment 6: The manuscript would benefit from more detailed comparisons with current literature, including deeper analysis of why the proposed method outperforms existing models.

Response 6: Thank you for this suggestion. To provide a deeper analysis, we have expanded the Discussion section (referencing p.28 486-493) with a more detailed comparison between DNABERT and CrisprBERT. We address the observation that CrisprBERT performed competitively on certain datasets, and we discuss that while large-scale pre-training provides a generalizable foundation, task-specific architectures like the BiLSTM in CrisprBERT are also critical. We conclude by proposing that a hybrid model combining the strengths of both could be a promising direction for future research.

Comment 7: While attention weight visualization is a good step toward interpretability, the biological relevance of the patterns identified needs further validation or expert annotation.

Response 7: This is an excellent point. Acknowledging the limitations of our initial approach, we have replaced the simple attention visualization with more quantitative and advanced interpretability methods: SHAP and Integrated Gradients. The results are now detailed in a new section, "3.4. Model Interpretation Unveils Predictive Mechanisms" (referencing p.22 383-446). Crucially, we have also expanded the Discussion (referencing p.28 459-485) to connect the patterns identified by these methods (e.g., the importance of H3K27ac, the two sequence hotspots) to recent biophysical studies on Cas9's R-loop formation and structural activation mechanisms, thereby directly addressing your comment on establishing "biological relevance."

Comment 8: The writing is generally clear, but some sections—especially the methodology—could be more concise and better organized with clearer subheadings.

Response 8: We agree. We have reorganized the "Materials and Methods" section by introducing a clearer hierarchical structure with more descriptive subheadings (e.g., 2.1. Datasets, 2.2. Epigenetic Feature Processing), which we believe has significantly improved its clarity and organization.

Responses to Reviewer #2

Comment 1: While the use of DNABERT and epigenetic features is compelling, the authors could consider more clearly articulating how their approach differs from previous BERT-based off-target prediction models in terms of integration strategy or architecture.

Response 1: Thank you for this comment. We have revised the manuscript to better articulate these differences. In the Introduction (referencing p.3 47-54), we now clarify that the primary distinction is the use of a pre-trained DNA foundation model rather than a model trained from scratch. Furthermore, the specific architecture for integrating epigenetic features, which employs a gating mechanism, is now detailed in section " 2.3.2. DNABERT-Epi for Multimodal Integration" and illustrated in Figure 1C (referencing p.9 148-160).

Comment 2: The rationale for choosing only H3K4me3, ATAC-seq, and H3K27ac is underexplored. Please explain why these three features were selected over others and discuss their individual contributions to performance.

Response 2: This is an important point that required clarification. We have addressed it in two ways. First, we now provide a clear rationale for our feature selection in section " 2.2. Epigenetic Feature Processing" (referencing p.6 101-106), which is based on the findings from the original dataset paper (Lazzarotto et al., 2020) that reported a significant enrichment of precisely these three marks. Second, we have performed a quantitative analysis of the individual contributions of these features using SHAP, with the results presented in section " 3.4.1. SHAP Reveals the Critical Role of H3K27ac" and Figure 5A, showing that H3K27ac is the most impactful feature (referencing p.23 389-416).

Comment 3: The model was tested primarily on CHANGE-seq and GUIDE-seq datasets. Would the approach generalize to other datasets or Cas variants? A brief discussion on applicability beyond Cas9 would strengthen the impact.

Response 3: We agree completely. To address this, we have significantly expanded our validation. First, we added five additional in cellula datasets, using four of them as completely independent test sets to rigorously evaluate the model's generalization performance. These results are presented in section " 3.1. Benchmarking Performance on Diverse Datasets" and Figure 3 (referencing p.16 273-319). Second, we now discuss the applicability to other Cas variants as a key limitation and area for future work in the Discussion section, with citations to relevant literature (referencing p.28 494-497).

Comment 4: The attention weight visualization is promising but limited to seed/non-seed distinctions. Including more quantitative or alternative interpretation techniques would enhance model transparency.

Response 4: Thank you for this constructive suggestion. We have replaced our original attention visualization with more quantitative and powerful interpretability techniques. We now use SHAP for epigenetic features and Integrated Gradients for sequence features. These new methods and their results are detailed in sections " 2.6. Model Interpretability Analysis" and "3.4. Model Interpretation Unveils Predictive Mechanisms", as well as in Figure 5 and Figure 6 (referencing p.22 383-446).

Comment 5: The ensemble method consistently outperforms individual models, but the contribution of each component remains unclear. An ablation of the ensemble composition would better justify its use.

Response 5: Thank you for this constructive suggestion. As suggested, we performed a leave-one-out analysis to ablate the ensemble composition. We have added a new section, " 3.3. Ensemble Model Benefits from Diverse Architectures", and a new figure, Figure 4, to present the results of this analysis, which quantifies how much each individual model contributes to the ensemble's overall performance (referencing p.21 358-381).

Comment 6: The authors are also encouraged to cite recent advances in sequence-based RNA/DNA function or modification prediction, such as https://doi.org/10.1021/acs.jcim.4c00546 and https://doi.org/10.1016/j.csbj.2025.02.029.

Response 6: We thank the reviewer for bringing these recent and relevant papers to our attention. We have reviewed both papers and have now cited them in the Introduction (referencing p.3 39-41) and Discussion (referencing p.29 519-523) sections to enrich the context of our work.

Point regarding Statistical Analysis (from reviewer checklist):

We sincerely thank the reviewer for highlighting the initial weaknesses in our statistical analysis. This was the most critical point of feedback, and we have taken significant steps to address it thoroughly to ensure the robustness of our conclusions.

In the revised manuscript, we have implemented a new, more rigorous statistical testing framework, as detailed in the new section " 2.5.3. Statistical Analysis" (referencing p.14 229-237). The key improvements are:

1. We now employ a two-sided Wilcoxon signed-rank test for all pairwise model comparisons, which is appropriate for our paired cross-validation design.

2. Most importantly, to address the issue of multiple comparisons, we now control the false discovery rate (FDR) using the Benjamini-Hochberg (BH) procedure. All p-values reported in the manuscript are now BH-adjusted p-values.

We are confident that these substantial revisions have corrected the deficiencies in our original analysis and now meet the standards of statistical rigor required.

We believe that these revisions have substantially strengthened the manuscript. We once again thank you for your time and constructive feedback.

Sincerely,

Kai Kimata

Graduate School of Natural Science and Technology, Kanazawa University, Kanazawa, Japan

tynkkaiosaka@stu.kanazawa-u.ac.jp

---

## [Decision Letter · Decision Letter 1]

16 Oct 2025

Improved CRISPR/Cas9 Off-target Prediction with DNABERT and Epigenetic Features

PONE-D-25-35489R1

Dear Dr. Satou,

We’re pleased to inform you that your manuscript has been judged scientifically suitable for publication and will be formally accepted for publication once it meets all outstanding technical requirements.

Kind regards,

Hodaka Fujii, M.D., Ph.D.

Academic Editor

PLOS ONE

Additional Editor Comments (optional):

Reviewers' comments:

Reviewer's Responses to Questions

**Comments to the Author**

Reviewer #1: All comments have been addressed

Reviewer #2: All comments have been addressed

2. Is the manuscript technically sound, and do the data support the conclusions?

Reviewer #1: Yes

Reviewer #2: Yes

3. Has the statistical analysis been performed appropriately and rigorously?

Reviewer #1: Yes

Reviewer #2: Yes

4. Have the authors made all data underlying the findings in their manuscript fully available?

Reviewer #1: (No Response)

Reviewer #2: Yes

5. Is the manuscript presented in an intelligible fashion and written in standard English?

Reviewer #1: Yes

Reviewer #2: Yes

Reviewer #1: The author has carefully and thoroughly addressed all the comments and suggestions provided by the reviewers, ensuring that each point has been considered and incorporated into the revised manuscript. Substantial improvements have been made to enhance the clarity, depth, and overall quality of the work. The revisions strengthen the scientific contribution and improve the manuscript’s readability, coherence, and impact. Given these significant enhancements and the author’s diligent efforts to meet the reviewers’ expectations, I believe the manuscript is now well-prepared for publication. I wish the author the very best of luck with the submission process.

Reviewer #2: (No Response)

**Do you want your identity to be public for this peer review?** For information about this choice, including consent withdrawal, please see our Privacy Policy

Reviewer #1: **Yes: ** salman Khan

Reviewer #2: No

---

## [Editor Report · Acceptance letter]

PONE-D-25-35489R1

PLOS ONE

Dear Dr. Satou,

I'm pleased to inform you that your manuscript has been deemed suitable for publication in PLOS ONE. Congratulations! Your manuscript is now being handed over to our production team.

Kind regards,

on behalf of

Dr. Hodaka Fujii

Academic Editor

PLOS ONE